# Learning Sparse Distributions using Iterative Hard Thresholding

**Jacky Y. Zhang**
Department of Computer Science
University of Illinois at Urbana-Champaign
yiboz@illinois.edu

**Rajiv Khanna**
Department of Statistics
University of California at Berkeley
rajivak@berkeley.edu

**Anastasios Kyrillidis**
Department of Computer Science
Rice University
rajivak@berkeley.edu

**Oluwasanmi Koyejo**
Department of Computer Science
University of Illinois at Urbana-Champaign
sanmi@illinois.edu

## Abstract

Iterative hard thresholding (IHT) is a projected gradient descent algorithm, known to achieve state of the art performance for a wide range of structured estimation problems, such as sparse inference. In this work, we consider IHT as a solution to the problem of learning sparse discrete distributions. We study the hardness of using IHT on the space of measures. As a practical alternative, we propose a greedy approximate projection which simultaneously captures appropriate notions of sparsity in distributions, while satisfying the simplex constraint, and investigate the convergence behavior of the resulting procedure in various settings. Our results show, both in theory and practice, that IHT can achieve state of the art results for learning sparse distributions.

## 1 Introduction

Probabilistic models provide a flexible approach for capturing uncertainty in real world processes, with a variety of applications which include latent variable models and density estimation, among others. Like other machine learning tools, probabilistic models can be enhanced by encouraging parsimony, as this captures useful inductive biases. In practice, this often improves the interpretability and generalization performance of the resulting models, and is particularly useful in applied settings with limited samples compared to the model degrees of freedom. One of the most effective parsimonious assumptions is sparsity. As such, learning sparse distributions is a problem of broad interest in machine learning, with many applications [1–7].

The majority of approaches for sparse probabilistic modeling have focused on the construction of appropriate priors based on inputs from domain experts. The technical challenges there involve the challenges of prior design and inference [3, 1, 8], including methods that are additionally designed to exploit special structures [5, 4, 7] . More recently, there has been an interest in studying these algorithmic approaches from an optimization perspective [9–11], with the goal of a deeper understanding and, in some cases, even suggesting improvements over previous methods [12, 13].In this work, we consider an optimization-based approach to learning sparse discrete distributions. Despite wide applicability, when compared to classical constrained optimization, there are limited studies that focus on the understanding, both in theory and in practice, of optimization methods over the space of probability densities, under sparsity constraints.

Our present work proposes and investigates the use of Iterative Hard Thresholding (IHT [14–18]) for the problem of sparse probabilistic estimation. IHT is an iterative algorithm that is well-studied in the

classical optimization literature. Further, there are known worst-case convergence guarantees and empirical studies [19, 20] that vouch for its performance. Our goal in this work is to investigate the convergence properties of IHT, when applied to probabilistic densities, and to evaluate its efficacy for learning sparse distributions.

However, transferring this algorithm from vector and matrix spaces to the space of measures is not straightforward. While several of the technical pieces –such as the existence of a variational derivative and normed structure– fall into place, the algorithm is an iterative one, that involves solving a projection subproblem in each iteration. We show that this subproblem is computationally hard in general, but provide an approximate procedure that we analyze under certain assumptions.

Our contributions in this work are algorithmic and theoretical, with proof of concept empirical evaluation. We briefly summarize our contributions below.

- We propose the use of classical IHT for learning sparse distributions, and show that the space of measures meets the structural requirements for IHT.
- We study in depth the hardness of the projection subproblem, showing that it is NP-hard, and no polynomial-time algorithm exists that can solve it with guarantees.
- Since the projection problem is solved in every iteration, we propose a simple greedy algorithm and provide sufficient theoretical conditions, under which the algorithm provably approximates the otherwise hard projection problem.
- We draw on techniques from classical optimization to provide convergence rates for the overall IHT algorithm: i.e., we study after how many iterations will the algorithm guarantee to be within some small $\epsilon$ of the true optimum.

In addition to our conceptual and theoretical results, we present empirical studies that support our claims.

## 2    Problem statement

**Preliminaries.** We use bold characters to denote vectors. Given a vector $\boldsymbol{v}$, we use $v_i$ to represent its $i$-th entry. We use calligraphic upper case letters to denote sets; *e.g.*, $\mathcal{S}$. With a slight abuse of notation, we will use lower case letters to denote probability distributions *e.g.*, $p, q$, as well as functions *e.g.*, $f$. The distinction from scalars will be apparent from the context; we usually append functions with parentheses to distinguish from scalars. We use upper case letters to denote functionals *i.e.*, functions that take as an input other functions *e.g.*, $F[p(\cdot)]$. We use $[n]$ to denote the set $\{1, 2, ...n\}$. Given a set of indices $\mathcal{S} \subset [n]$, we denote the cardinality of $\mathcal{S}$ as $|\mathcal{S}|$. Given a vector $\boldsymbol{x}$, we denote its support set *i.e.*, the set of non-zero entries, as $\text{supp}(\boldsymbol{x})$. We use $\mathbb{P}\{e\}$ to denote the probability of event $e$.

Let $\mathcal{P}$ denote the set of discrete $n$-dimensional probability densities on an $n$-dimensional domain $\mathcal{X}$ :

$$\mathcal{P} = \left\{ p(\cdot) \ : \mathcal{X} \to \mathbb{R}_+ \ \mid \ \sum_{\boldsymbol{x} \in \mathcal{X}} p(\boldsymbol{x}) = 1 \right\}.$$

Let $\mathcal{S} \subset [n]$ denote a support set where $|\mathcal{S}| = k < n$. Let $\mathcal{X}_{\mathcal{S}} \subset \mathcal{X}$ denote the set of variables with support $\mathcal{S}$, *i.e.*,

$$\mathcal{X}_{\mathcal{S}} = \big\{ \boldsymbol{x} \in \mathcal{X} \ \mid \ \text{supp}(\boldsymbol{x}) \subseteq \mathcal{S} \big\}.$$

The set of domain restricted densities, denoted by $\mathcal{P}_{\mathcal{S}}$, is the set of probability density functions supported on $\mathcal{X}_{\mathcal{S}}$; *i.e.*,

$$\mathcal{P}_{\mathcal{S}} = \{q(\cdot) \in \mathcal{P} \mid \forall \boldsymbol{x} \notin \mathcal{X}_{\mathcal{S}}, \ q(\boldsymbol{x}) = 0\} .$$

Inversely, we denote the support of a domain restricted density $q(\cdot) \in \mathcal{P}_{\mathcal{S}}$ as $\text{supp}(q) = \mathcal{S}$. Next, we define the notion of sparse distributions.

**Definition 1** (**Distribution Sparsity [5]**)**.** *Let $\mathcal{D}_k = \cup_{|\mathcal{S}| \le k} \mathcal{P}_{\mathcal{S}} \subseteq \mathcal{P}$ i.e., the union of all possible $k$-sparse support domain restricted densities. We say that $p(\cdot)$ is $k$-sparse if $p(\cdot) \in \mathcal{D}_k$.*

Note that while each component $\mathcal{P}_{\mathcal{S}}$ is a convex set, the union $\mathcal{D}_k$ is not. To see this, consider the convex combination of two $k$-sparse distributions $p_1$ and $p_2$ with disjoint supports $\mathcal{S}_1$ and $\mathcal{S}_2$ respectively. In general, the convex combination $\alpha p_1(\cdot) + (1 - \alpha)p_2(\cdot); \ 0 < \alpha < 1$, has larger support; i.e., $|\mathcal{S}_1 \cup \mathcal{S}_2| > k$. As an aside, we note that unlike the vector case, its is straightforward

to construct multiple definitions of distribution sparsity. For instance, another reasonable definition is via the set $\mathcal{D}'_k = \{p(\cdot) \in \mathcal{P} \mid p(\boldsymbol{x}) = 0 \text{ for all } \|\boldsymbol{x}\|_0 > k\}$; i.e., distributions that assign zero probability mass to non-$k$-sparse vectors. Interestingly, $\mathcal{D}_k \subset \mathcal{D}'_k \subset \mathcal{P}$ in general, as any of the distributions in $\mathcal{D}_k$ must has a support with size less than $k$, which is not necessary for distributions in $\mathcal{D}'_k$. Motivated by prior work [5], we use Definition 1 in this work.

**Vector sparsity.** While the proposed framework is developed for a specialized notion of sparsity i.e. along the dimensions of a multivariate discrete distribution, it is also applicable to alternative notions of distribution sparsity. One common setting is sparsity of the distribution itself $p(\cdot)$ when represented as a vector e.g. sparsifying the number of valid states of a univariate distribution such as a histogram. We outline how our framework can be applied to this setting in the Appendix A.

**Problem setting.** In this work, we focus on studying sparsity for the case of discrete densities. In particular, $\mathcal{X} \subset \mathbb{Z}^n$; i.e., $\boldsymbol{x}$ is an integer such that:
$$\mathcal{X} = \{\boldsymbol{x} \in \mathbb{Z}^n \mid \forall i \in [n], 0 \le x_i \le m - 1\},$$
where $m$ is an integer. Therefore, $\boldsymbol{x}$ has $m^n$ valid positions. In other words, if we denote $X$ as a random variable from that distribution, then $X \in \mathcal{X}$ has $m^n$ possible values, and $\mathbb{P}\{X = \boldsymbol{x}\} = p(\boldsymbol{x})$.

Given a cost functional over distributions $F[\cdot] : \mathcal{P} \to \mathbb{R}$, we are interested in the following optimization criterion:
$$\min_q \quad F[q] \quad \text{subject to} \quad q \in \mathcal{D}_k, \tag{1}$$
where $\mathcal{D}_k = \cup_{\mathcal{S}:|\mathcal{S}| \le k} \mathcal{P}_{\mathcal{S}} \subseteq \mathcal{P}$ is the $k$-sparsity constraint, as in Definition 1. In words, we are interested in finding a *distribution*, denoted as $q(\cdot)$, that "lives" in the $k$-sparse set of distributions, and minimizes the cost functional $F[\cdot]$. This is similar to classical *sparse optimization* problems in literature [21–24], but there are fundamental difficulties, both in theory and in practice, that require a different approach than standard iterative hard thresholding algorithms [14–18].

We assume that the objective $F[\cdot]$ is a *convex* functional over distributions.

**Definition 2** (**Convexity of** $F[\cdot]$). *The functional* $F[\cdot] : \mathcal{P} \to \mathbb{R}$ *is convex if:*
$$F[\theta q(\cdot) + (1 - \theta)p(\cdot)] \le \theta F[q(\cdot)] + (1 - \theta)F[p(\cdot)],$$
*for all* $q(\cdot),\ p(\cdot) \in \mathcal{P}$ *and* $\theta \in [0, 1]$.

Observe that, while $F[\cdot]$ is a convex functional, and $\mathcal{P}$ and $\mathcal{P}_{\mathcal{S}}$ are convex sets, $\mathcal{D}_k$ is not a convex set. Hence, the optimization problem (1) is not a convex program.

Following the projected gradient descent approach, we require definitions of the gradient over $F[\cdot]$, as well as definitions of the projection.

**Definition 3** (**Variational Derivative** [25]). *The variational derivative of* $F[\cdot] : \mathcal{P} \to \mathbb{R}$ *is a function, denoted as* $\frac{\delta F}{\delta q}(\cdot) : \mathcal{X} \to \mathbb{R}$, *and satisfies:*
$$\sum_{\mathcal{X}} \frac{\delta F}{\delta q}(\boldsymbol{x})\phi(\boldsymbol{x}) = \frac{\partial F[q + \epsilon\phi]}{\partial \epsilon}\bigg|_{\epsilon=0}$$
*where* $\phi : \mathcal{X} \to \mathbb{R}$ *is an arbitrary function.*

**Definition 4** (**First-order Convexity**). *The functional* $F[\cdot] : \mathcal{P} \to \mathbb{R}$ *is convex if:*
$$F[q(\cdot)] \ge F[p(\cdot)] + \left\langle \frac{\delta F}{\delta p}(\cdot), q(\cdot) - p(\cdot) \right\rangle$$
*for all* $q(\cdot),\ p(\cdot) \in \mathcal{P}$.

Here, we use the standard inner product for two densities: $\langle q(\cdot), p(\cdot)\rangle = \int_x q(x)p(x)$, or $\langle q(\cdot), p(\cdot)\rangle = \sum_x q(x)p(x)$ in the discrete setting.

## 3  Algorithms

Recall that our goal is to solve the optimization problem (1). A natural way to solve it in an iterative fashion is using *projected gradient descent*, where the projection step is over the set of sparse distributions $\mathcal{D}_k$. This analogy makes the connection to iterative hard thresholding (IHT) algorithms, where the iterative recursion is:
$$p_{t+1}(\cdot) = \Pi_{\mathcal{D}_k}\left(p_t(\cdot) - \mu\frac{\delta F}{\delta p_t}(\cdot)\right),$$
where $p_t(\cdot)$ denotes the current iterate, and $\Pi_{\mathcal{D}_k}(\cdot)$ denotes, in an abstract sense, the projection of the distribution function to the set of sparse distribution functions.

The consequent steps are analogous to those of regular IHT: given an initialization point, we iteratively *i)* compute the gradient, *ii)* perform the gradient step with step size $\mu$, *iii)* ensure the computed approximate solution satisfies our constraint in each iteration by projecting to $\mathcal{D}_k$.

### 3.1 Projection onto $\mathcal{D}_k$

Consider the projection step with respect to the $\ell_2$-norm i.e.

---

**Algorithm 1** Distribution IHT

---

1: **Input:** $F[\cdot] : \mathcal{P} \to \mathbb{R}$, $k \in \mathbb{Z}_+$. number of iters $T$, $p_0(\cdot) \in \mathcal{D}_k$, $\mu$. **Output:** $p_T \in \mathcal{D}_k$
2: $t \leftarrow 0$
3: **while** $t < T$ **do**
4: $\quad q_{t+1}(\cdot) = p_t(\cdot) - \mu \frac{\delta F}{\delta p_t}(\cdot)$
5: $\quad p_{t+1}(\cdot) = \Pi_{\mathcal{D}_k}(q_{t+1})$
6: **end while**
7: **return** $p_T(\cdot)$

---

$$\Pi_{\mathcal{D}_k}(p(\cdot)) := \underset{q(\cdot) \in \mathcal{D}_k}{\arg\min} \|q(\cdot) - p(\cdot)\|_2^2, \tag{2}$$

where the $\ell_2$-norm is defined by the aforementioned inner product $\langle q(\cdot), p(\cdot) \rangle = \sum_{\boldsymbol{x}} q(\boldsymbol{x})p(\boldsymbol{x})$. The set $\mathcal{D}_k = \cup_{|\mathcal{S}| \leq k} \mathcal{P}_{\mathcal{S}}$ is a union of $\binom{n}{k} = O(n^k)$ sparse sets $\mathcal{P}_S$ of different supports. Thus, if we denote $\mathrm{T}_{\mathrm{proj}}$ as the time to compute $\Pi_{\mathcal{P}_S}(p(\cdot))$, then we need $O(n^k \cdot \mathrm{T}_{\mathrm{proj}})$ time for $\mathcal{D}_k$ projection using naive enumeration. One may reasonably conjecture the existence of more efficient implementations of the exact projection in (2), e.g., in polynomial time. In the following, we show that this is not the case.

### 3.2 On the tractability of sparse distribution $\ell_2$-norm projection

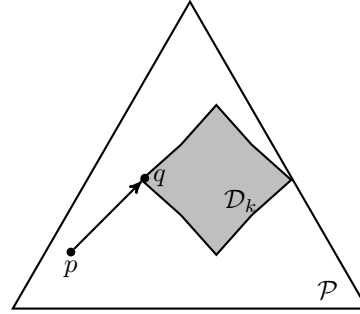

Figure 1: Illustration of projection onto $\mathcal{D}_k$, with $q = \Pi_{\mathcal{D}_k}(p)$.

The projection (2) is iteratively solved in IHT (step 5 in Algorithm 1). Thus, for the algorithm to be practical, it is important to study the tractability of the projection step. The combinatorial nature of $\mathcal{D}_k$ hints that this might not be the case.

**Theorem 1.** *The sparse distribution $\ell_2$-norm projection problem* (2) *is NP-hard.*

*Sketch of proof:* We show that the subset selection problem [26] can be reduced to the $\ell_2$-norm projection problem. The complete proof is provided in the supplementary material.

As an alternative route, NP-hard problems can be often tackled sufficiently, by using approximate methods. However, the following theorem states that the sparsity constrained optimization problem in (2) is hard even to approximate, in the sense that no deterministic approximation algorithm exists that solves it in polynomial time.

**Theorem 2.** *There exists no deterministic algorithm that can provide a constant factor approximation for the sparse distribution $\ell_2$-norm projection problem in polynomial time. Formally, for given $q : \mathcal{X} \to \mathbb{R}$ with $\mathcal{X} \in \mathbb{R}^n$, let $p^\star(\cdot)$ be the optimal $\ell_2$-norm projection onto $\mathcal{D}_k$, and let $\widehat{p}(\cdot)$ be the solution found by any algorithm that operates in $O(\mathrm{poly}(n))$ time. Then, we can design problem instances, where the approximation ratio:*

$$\varphi = \frac{\|q(\cdot) - \widehat{p}(\cdot)\|_2^2}{\|q(\cdot) - p^\star(\cdot)\|_2^2} - 1,$$

*cannot be bounded.*

The proof of the theorem is provided in the supplementary material. Through Theorems 1 and 2, we have shown that the distribution sparse $\ell_2$-norm projection problem is hard, and thus the applicability of IHT on the space of densities seems not to be well-established to be practical. This may be surprising, in light of results in a variety of domains where it is known to be effective. For example, in case of vectors, a simple $O(n)$ selection algorithm solves the projection problem *optimally* [27]. Similarly, on the space of matrices for low rank IHT, the projection onto the top-$k$ ranks is optimally solved by an SVD [28].

## 3.3 A greedy approximation

In contrast to the results of Theorems 1 and 2, we have observed that a simple greedy support selection seems effective in practice. Thus, we simply consider replacing exact projection to $\mathcal{D}_k$ by greedy selection.

Consider Algorithm 2 when the input is not necessarily a distribution, *i.e.*, $\sum_{\boldsymbol{x}\in\mathcal{X}} q(\boldsymbol{x}) \neq 1$. The key procedure of the projection is line 5, where the inner $\min(\cdot)$ is the projection of $q(\cdot)$ on a set of domain restricted densities. Let

---

**Algorithm 2** Greedy Sparse Projection (GSProj)

1: **Input:** $n$-dimensional function $q : \mathcal{X} \to \mathbb{R}$ and sparsity level $k$.
2: **Output:** A distribution $p(\cdot) \in \mathcal{D}_k$
3: $\mathcal{S} := \emptyset$
4: **while** $|\mathcal{S}| < k$ **do**
5: $\quad j \in \arg\min_{i\in[n]\setminus\mathcal{S}} \left\{\min_{p\in\mathcal{P}_{\mathcal{S}\cup i}} \|p(\cdot) - q(\cdot)\|_2^2\right\}$
6: $\quad \mathcal{S} := \mathcal{S} \cup j$
7: **end while**
8: **return** $\arg\min_{p\in\mathcal{P}_{\mathcal{S}}} \|p(\cdot) - q(\cdot)\|_2^2$

---

$\widehat{p}(\cdot)$ denote this projection, *i.e.*, $\widehat{p}(\cdot) = \arg\min_{p(\cdot)\in\mathcal{P}_{\mathcal{S}}} \|p(\cdot) - q(\cdot)\|_2^2$. Since, by definition $\widehat{p}(\boldsymbol{x}) = 0$ for any $\boldsymbol{x} \notin \mathcal{X}_{\mathcal{S}}$, we only need to calculate $\widehat{p}(\boldsymbol{x})$ where $\boldsymbol{x} \in \mathcal{X}_{\mathcal{S}}$, and this can be reformulated as:

$$\arg\min_{p(\cdot)} \sum_{\boldsymbol{x}\in\mathcal{X}_{\mathcal{S}}} (p(\boldsymbol{x}) - q(\boldsymbol{x}))^2 \quad \text{s.t.} \quad \sum_{\boldsymbol{x}\in\mathcal{X}_{\mathcal{S}}} p(\boldsymbol{x}) = 1 \quad \text{and} \quad \forall_{\boldsymbol{x}\in\mathcal{X}_{\mathcal{S}}} p(\boldsymbol{x}) \geq 0,$$

which is essentially $\ell_2$-norm projection onto a simplex $\{p(\boldsymbol{x}) \mid \sum_{\boldsymbol{x}\in\mathcal{X}_{\mathcal{S}}} p(\boldsymbol{x}) = 1, \forall_{\boldsymbol{x}\in\mathcal{X}_{\mathcal{S}}} p(\boldsymbol{x}) \geq 0\}$. This $\ell_2$-norm projection onto the simplex can be solved efficiently and easily (See [29]).

When $p(\cdot)$ is a distribution, we can analytically compute its projection on any support restricted domain. Given support $\mathcal{S}$, the exact projection of a distribution $p(\cdot)$ onto $\mathcal{P}_{\mathcal{S}}$ is:

$$\arg\min_{q\in\mathcal{P}_{\mathcal{S}}} \|q(\cdot) - p(\cdot)\|_2^2. \tag{3}$$

In our setting, the above problem can be written as

$$\arg\min_{q\in\mathcal{P}_{\mathcal{S}}} \|q(\cdot) - p(\cdot)\|_2^2 = \arg\min_{q\in\mathcal{P}_{\mathcal{S}}} \langle q(\cdot) - p(\cdot), q(\cdot) - p(\cdot)\rangle = \arg\min_{q\in\mathcal{P}_{\mathcal{S}}} \sum_{\boldsymbol{x}\in\mathcal{X}} (q(\boldsymbol{x}) - p(\boldsymbol{x}))^2$$

$$= \arg\min_{q\in\mathcal{P}_{\mathcal{S}}} \sum_{\boldsymbol{x}\in\mathcal{X}_{\mathcal{S}}} (q(\boldsymbol{x}) - p(\boldsymbol{x}))^2 + \sum_{\boldsymbol{x}\in\mathcal{X}, \boldsymbol{x}\notin\mathcal{X}_{\mathcal{S}}} p(\boldsymbol{x})^2.$$

The last equation is due to definition of $\mathcal{P}_{\mathcal{S}}$ and $\mathcal{X}_{\mathcal{S}}$. Since $p(\cdot)$ is constant, we can eliminate the last term. Further, since $q \in \mathcal{P}_{\mathcal{S}}$, we have that $q(\boldsymbol{x}) = 0$ for every $\boldsymbol{x} \notin \mathcal{X}_{\mathcal{S}}$. The resulting problem is:

$$\arg\min_{q\in\mathcal{P}_{\mathcal{S}}} \sum_{\boldsymbol{x}\in\mathcal{X}_{\mathcal{S}}} (q(\boldsymbol{x}) - p(\boldsymbol{x}))^2 \quad \text{s.t.} \quad \sum_{\boldsymbol{x}\in\mathcal{X}_{\mathcal{S}}} q(\boldsymbol{x}) = 1. \tag{4}$$

Denote $\sum_{\boldsymbol{x}\in\mathcal{X}_{\mathcal{S}}} p(\boldsymbol{x}) = C \leq 1$. Applying the Quadratic Mean-Arithmetic Mean inequality to equation (4), we have:

$$\sum_{\boldsymbol{x}\in\mathcal{X}_{\mathcal{S}}} (q(\boldsymbol{x}) - p(\boldsymbol{x}))^2 \geq (1 - C)^2 / |\mathcal{X}_{\mathcal{S}}| \quad \text{s.t.} \quad \sum_{\boldsymbol{x}\in\mathcal{X}_{\mathcal{S}}} q(\boldsymbol{x}) = 1$$

The equality can be achieved when $q(\boldsymbol{x}) - p(\boldsymbol{x})$ is the same for every $\boldsymbol{x} \in \mathcal{X}_{\mathcal{S}}$. Therefore we have the optimal solution to Problem (3):

$$q_{\mathcal{S}}^{\star}(\boldsymbol{x}) = \begin{cases} p(\boldsymbol{x}) + \frac{1-C}{|\mathcal{X}_{\mathcal{S}}|}, & \boldsymbol{x} \in \mathcal{X}_{\mathcal{S}} \\ 0, & \boldsymbol{x} \notin \mathcal{X}_{\mathcal{S}} \end{cases}$$

**Computational complexity.** The time we need to solve Problem (3) is $O(|\mathcal{X}_{\mathcal{S}}|)$, i.e. the time to compute $C$. However, to compute the norm $\|q(\cdot) - p(\cdot)\|_2^2$ we still need $O(|\mathcal{X}|)$ time, as $p(\boldsymbol{x})$ is not necessarily zero at any $\boldsymbol{x} \in \mathcal{X}$. As a result, we need $O(n^k(|\mathcal{X}| + |\mathcal{X}_{\mathcal{S}}|))$ time to enumerate for an optimal solution of the $\ell_2$-norm projection. If we consider the integer lattice $\mathcal{X}$, as stated in the problem setting, then $|\mathcal{X}| = m^n$ and $|\mathcal{X}_{\mathcal{S}}| = m^k$, rendering the time complexity $O(n^k m^n)$. However, Algorithm 2 has much lower time complexity. In each iteration, the greedy method selects an element to put into $\mathcal{S}$ that maximize the gain, which requires $k$ iterations. It need not to consider the exact $\ell_2$-norm $\|q(\cdot) - p(\cdot)\|_2^2$ in each iteration, only the increment for each $e$ from $n$ options. To compute the increment, no more than $|\mathcal{X}_{\mathcal{S}}|$ terms are added, which requires compute of $O(|\mathcal{X}_{\mathcal{S}}|)$ time complexity. All together, the greedy method requires $O(k|\mathcal{X}_{\mathcal{S}}|)$ time to operate, or $O(nkm^k)$ in our integer lattice setting, which is far less that the enumeration method's $O(n^k m^n)$.

## 3.4 When Greedy is Good

We have shown in the proof of Theorem 2 that there always exist extreme examples that are hard to solve. Thus, in the most general sense, and without further assumptions, one can find pathological cases which make the problem hard. However, we find that the greedy approach works well empirically. In this section, we consider sufficient conditions for tractability of the problem. Our conditions boil down to structural assumptions on $F[\cdot]$ which match standard assumptions in the literature.

To build further intuition, consider line 4 in Algorithm 1, where the parameter passed to the greedy method is $q(\cdot) = p(\cdot) - \mu \frac{\delta F}{\delta p}(\cdot)$, and $p(\cdot)$ is already a $k$-sparse distribution. Denote the support of $p(\cdot)$ as $\mathcal{S}$; we can see that $|\mathcal{S}| \leq k$. Therefore, that $q(\cdot)$ is close to $k$-sparse when the step size $\mu$ is small. Thus, while the general problem (2) may be a lot harder, there is reason to conjecture that under certain conditions, a simple greedy algorithm performs well. Next, we state these assumptions formally.

**Assumption 1** (**Strong Convexity/Smoothness**). *The objective $F[\cdot]$ satisfies Strong Convexity/Smoothness with respect to $\alpha$ and $\beta$ if:*

$$\frac{\alpha}{2}\|p_1(\cdot) - p_2(\cdot)\|_2^2 \leq F[p_1(\cdot)] - F[p_2(\cdot)] - \left\langle \frac{\delta F}{\delta p_1}(\cdot), p_2(\cdot) - p_1(\cdot) \right\rangle \leq \frac{\beta}{2}\|p_1(\cdot) - p_2(\cdot)\|_2^2$$

For the sake of simplicity in exposition, we have assumed strong convexity to hold over the entire domain (which can be a restrictive assumption). As will be clear from the proof analysis, this assumption can easily be tightened to a restricted strong convexity assumption; see, e.g., [30]. This detail is left for a longer version of this manuscript.

**Assumption 2** (**Lipschitz Condition**). *The functional $F : \mathcal{P} \rightarrow \mathbb{R}$ satisfies the Lipschitz condition with respect to $L$, in $k$-sparse domain $\mathcal{D}_k$ is*

$$|F[p_1(\cdot)] - F[p_2(\cdot)]| \leq L\|p_1(\cdot) - p_2(\cdot)\|_2$$

*This assumption implies that*

$$\left\|\frac{\delta F}{\delta p}(\cdot)\right\|_2 \leq L.$$

Using the strong convexity, smoothness, and Lipschitz assumptions, we are able to provide analysis for when greedy works well. This is encapsulated in Theorem 3.

**Theorem 3.** *Given $n$-dimensional function $q(\cdot) = p(\cdot) - \mu \frac{\delta F}{\delta p}(\cdot)$, where $p(\cdot)$ is an $n$-dimensional $k$-sparse distribution and $supp(p(\cdot)) = \mathcal{S}'$, Algorithm 2 finds the optimal projection to domain $\mathcal{P}_{\mathcal{S}'}$ if $F[\cdot]$ satisfies Assumption 2, $\mu$ is sufficiently small and there are enough positions $\boldsymbol{x} \in \mathcal{X}_{\mathcal{S}'}$ where $p(\boldsymbol{x}) > 0$, i.e., satisfies inequality (6) and inequality (9).*

## 3.5 Convergence Analysis

Next, we analyze the convergence of the overall Algorithm 1 with greedy projections. While Theorem 3 provides sufficient conditions for exact projection using the greedy approach, in practice due to computational precision issues and/or violation of the stated assumptions, the solution may not provide an exact projection. Thus, it is prudent to assume that the inner projection subproblem is solved within some approximation as quantified in the following.

**Definition 5.** *Approximate $\ell_2$-norm projection. We define $\widehat{\Pi}_{\mathcal{D}_k}(\cdot)$ as the approximate projection onto sparsity domain and distribution space, with approximation parameter, $\phi$, as:*

$$\left\|p(\cdot) - \widehat{\Pi}_{\mathcal{D}_k}(p(\cdot))\right\|_2^2 \leq (1 + \phi)\|p(\cdot) - \Pi_{\mathcal{D}_k}(p(\cdot))\|_2^2$$

Next, we present our main convergence theorem.

**Theorem 4.** *Suppose $F$ satisfies assumptions 1 and 2. Furthermore, assume that the projection step in Algorithm 1 is solved $\phi$-approximately. Let the step size $\mu = 1/\beta$, and $\|p_0(\cdot) - p^\star(\cdot)\|_2 \leq L/(2\alpha)$. Then if $\frac{\beta}{\alpha} \in (2 - \frac{1}{1+\phi}, 2)$, IHT (Algorithm 1) with $T \geq \log_\eta \frac{\epsilon}{F[p_0(\cdot)] - F[p^\star(\cdot)] - c}$ iterations achieves $F[p_T(\cdot)] \leq F[p^\star(\cdot)] + c + \epsilon$, where $\eta = 1 - (1 + \phi)(2 - \beta/\alpha)$ and $c = \frac{(\phi/(2\beta) + (1+\phi)(\beta-\alpha)/(2\alpha^2))L^2}{(1+\phi)(2 - \beta/\alpha)}$.*

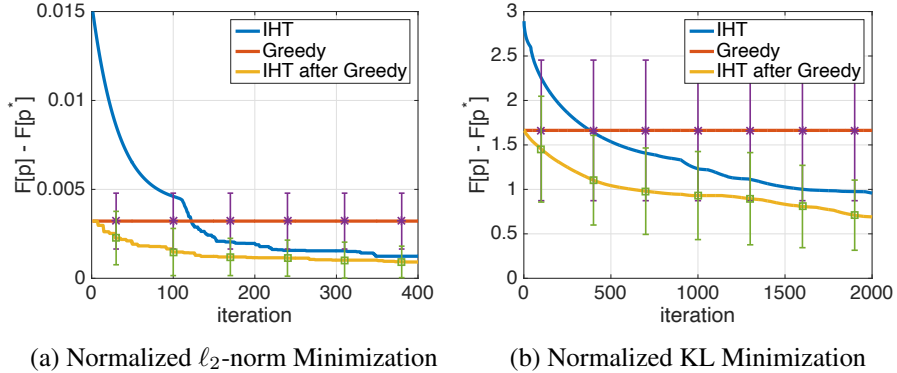

| (a) Normalized $\ell_2$-norm Minimization | (b) Normalized KL Minimization |

Figure 2: Simulated Experiments

# 4   Experiments

We evaluate our algorithm on different convex objectives, namely, $\ell_2$-norm distance and KL divergence. As mentioned before, there are no theoretically guaranteed algorithms for $\ell_2$-norm distance minimization under sparsity constraint. To investigate optimality of the algorithms, we consider simulated experiments of sufficiently small size that the global optimal can be exhaustively enumerated.

---
**Algorithm 3** Greedy Selection
---
1: **Input:** $F[\cdot] : \mathcal{P} \to \mathbb{R}$, $k \in \mathbb{Z}_+$. **Output:** $p_T \in \mathcal{D}_k$
2: $\mathcal{S} := \emptyset$
3: **while** $|\mathcal{S}| < k$ **do**
4:    $j \in \arg\min_{i \in [n] \setminus \mathcal{S}} \{\min_{p \in \mathcal{P}_{\mathcal{S} \cup i}} F[p(\cdot)]\}$
5:    $\mathcal{S} := \mathcal{S} \cup j$
6: **end while**
7: **return** $\arg\min_{p \in \mathcal{P}_{\mathcal{S}}} F[p(\cdot)]$
---

**IHT implementation details.** For IHT, the step size is chosen by a simple strategy: given an initial step size, we double the step size when IHT is trapped in local optima, and return to the initial step size after escaping. We return the algorithm along the entire solution path.

**Baseline: Forward Greedy Selection.** Unfortunately, we are unaware of optimization algorithms for sparse probability estimation with general losses. As as a simple baseline, we consider greedy selection wrt. the objective. This is equivalent to Algorithm 3. For certain special cases e.g. KL objective, Algorithm 3 can be applied efficiently and is effective in practice [5].

## 4.1   Simulated Data

We set dimension $n = 15$, number of entries $m = 2$, sparsity level $k = 7$. That is, $\mathcal{X} = \{0, 1\}^{15}$ is a 15-dimensional binary vector space, with cardinality $|\mathcal{X}| = 2^{15} = 32768$. The distribution $p : \mathcal{X} \to [0, 1]$ satisfies $\sum_{\boldsymbol{x} \in \mathcal{X}} p(\boldsymbol{x}) = 1$. The sparsity constraint is designed to fix a support $\mathcal{S} : |\mathcal{S}| \leq 7$, such that for any $\boldsymbol{x} : p(\boldsymbol{x}) > 0$ has $\text{supp}(\boldsymbol{x}) = \mathcal{S}$. Thus, the optimal solution is requires enumerating $\binom{15}{7} = 6435$ possible supports.

The $\ell_2$-norm minimization objective is $F[p(\cdot)] = \|p(\cdot) - q(\cdot)\|_2^2$ where $q(\cdot)$ is a distribution generated by randomly choosing 50 positions $x_1 \cdots x_{50} \in \mathcal{X}$ to assign random real numbers $c_1 \cdots c_{50}$ : $\sum_{i=1}^{50} c_i = 1$ and the other positions are assigned to 0, i.e., $q(x_i) = c_i$ for $i \in [50]$, and $q(\boldsymbol{x}) = 0$ otherwise. Initial step size $\mu = 0.008$. Results are shown in Figure 2 (a). For the KL divergence objective, it is $F[p(\cdot)] = KL(p(\cdot)\|q(\cdot)) = \sum_{\boldsymbol{x} \in \mathcal{X}} p(\boldsymbol{x}) \log \frac{p(\boldsymbol{x})}{q(\boldsymbol{x})}$, where $q(\cdot)$ is a random distribution generated similar to the $q(\cdot)$ in $\ell_2$-norm objective. The only difference is that $q(\boldsymbol{x})$ can not be zero as it would render the KL undefined. For simulated experiments, we use the optimum to normalize the objective function as $\tilde{F}[p] = F[p] - F[p^\star]$, so that at the optimum $\tilde{F}[p^\star] = 0$.

Three algorithms are compared in each experiment, i.e., IHT, Greedy and IHT after Greedy. While IHT starts randomly, IHT after Greedy is initialized by the result of Greedy. In each run, the distribution $q(\cdot)$ and the starting distribution for IHT $p_0(\cdot)$ are randomly generated. Each of the experiments are run 20 times. Results are presented in showing the mean and standard deviation of Greedy and IHT after Greedy. The standard deviation of IHT is similar to that of IHT after Greedy.

We use the $\ell_2$-norm greedy projection in IHT in both experiments. Interestingly, this not only outperforms the $\ell_2$-norm greedy projection itself (Figure 2 (a)), but also outperforms Greedy on the KL objective (Figure 2 (b)), where [5] suggests provably good performance. In particularly, while the performance of Greedy can fluctuate severely, IHT (after Greedy) is stable in obtaining good results. Note that low variance is especially desirable when the algorithm is only applied a few times to save computation, as in large discrete optimization problems.

## 4.2 Benchmark Data

**Distribution Compression / Compressed sensing.** We apply our IHT to the task of expectation-preserving distribution compression, useful for efficiently storing large probability tables. Given a distribution $p(\cdot)$, our goal is to construct a sparse approximation $q(\cdot)$, such that $q(\cdot)$ approximately preserves expectations with respect to $p(\cdot)$. Interestingly, this model compression problem is equivalent to compressed sensing, but with the distributional constraints. Specifically, our goal is to find $q$ which minimizes $||Aq - Ap||_2^2$ subject to a $k$-sparsity constraint on $q$. The model is evaluated with respect to moment reconstruction $||Bq - Bp||_2^2$ for a new "sensing" matrix $B$. Our experiments use real data from the Texas hospital discharge public use dataset. IHT is compared to post-precessed Lasso and Random. Lasso ignores the simplex constraints during optimization, then projects the results to the simplex, while Random is a naïve baseline of random distributions. Figure 3(a) shows that IHT significantly outperforms baselines. Additional details are provided in Appendix H due to limited space.

**Dataset compression.** We study representative prototype selection for the Digits data [31]. Prototypes are representative examples chosen from the data in order to achieve dataset compression. Our optimization objective is the Maximum Mean Discrepancy (MMD) between the discrete data distribution and the sparse data distribution representing the selected samples. We evaluate performance using the prototype nearest neighbor classification error on a test dataset. We compare two forward selection greedy variants (Local Greedy and Global Greedy) proposed by [32] and the means algorithm (labeled as PS) proposed by [33], both state of the art. The results are presented in Figure 3(b) showing that IHT outperforms all baselines. Additional experimental details are provided in Appendix H due to limited space.

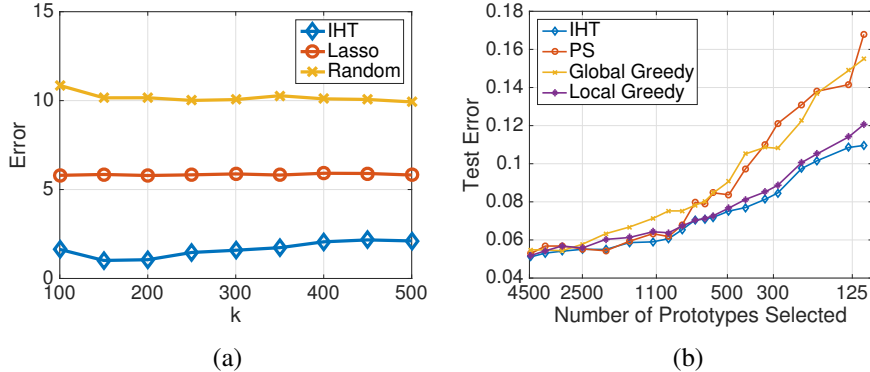

(a)                                        (b)

Figure 3: (a) Compression / Compressed sensing. Test Error at varying sparsity $k$. (b) Dataset Compression. Test Classification error of prototype nearest neighbor classifier

## 5 Conclusion and Future Work

In this work, we proposed the use of IHT for learning discrete sparse distributions. We study several theoretical properties of the algorithm from an optimization viewpoint, and propose practical solutions to solve otherwise hard problems. There are several possible future directions of research. We have analyzed discrete distributions with sparsity constraints. The obvious extensions are to the space of continuous measures and structured sparsity constraints. Is there a bigger class of constraints for which the a tractable projection algorithm exists? Can we improve the sufficient conditions under which projections are provably close to the optimum projection? Finally, more in-depth empirical studies compared to other state of the art algorithms should be very interesting and useful to the community.

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
