[Supplementary Material · distIHT-supp.pdf]

# A Vector-Sparsity for Distributions

While our framework is developed for sparsity along the dimensions of a multivariate discrete distribution, it is easily extended to alternative notions of sparsity. One common setting is where we are interested in sparsity of the distribution $p(\cdot)$ when represented as a vector $\mathbf{p}$ e.g. sparsifying the number of valid states of a univariate distribution such as a histogram. In our setting, this can be solved by constructing a binary vector $Z \in \mathcal{Z} = \{0, 1\}^{|\mathcal{X}|}$, where each dimension of $Z$ indexes one of the possible states of $X$. As each state of $X$ is associated with an element of the vector $\mathbf{p}$, state restrictions imply vector sparsity of $\mathbf{p}$. For example $\mathbf{z} = [1, 1, \ldots, 1, 0]$ implies that $X$ can take all states apart from the last one, $\mathbf{z} = [1, 1, 0, \ldots, 0]$ implies that only the first two states are valid, and so on. Setting $\mathbb{P}(Z = \mathbf{z}) \propto \mathbb{P}(X \in \{\text{states indexed by } \mathbf{z}\}) \propto \mathbf{z}^\top \mathbf{p}$ completes the transformation. In summary, dimension-wise sparsity of $\mathbf{z}$ corresponds to restrictons on the support of $X$, which equivalently corresponds to sparsification of the vector probability $\mathbf{p}$.

# B Connection between Variational Derivative and Gradient

As random variable has discrete space $\mathcal{X}$, we can use a vector to store probability of every $\boldsymbol{x} \in \mathcal{X}$ of a distribution $q(\cdot) : \mathcal{X} \to [0, 1]$. That is, the vector serves as an oracle of $q(\cdot)$. We denote the vector as $\widehat{\boldsymbol{q}} \in [0, 1]^{|\mathcal{X}|}$ to distinguish it from the original $q(\cdot) \in \mathcal{P}$.

We define a bijection map $\Phi(\cdot) : \mathcal{X} \to [|\mathcal{X}|]$, and let $\forall \boldsymbol{x} \in \mathcal{X}$, we have $q(\boldsymbol{x}) = \widehat{q}_{\Phi(\boldsymbol{x})}$, where $\widehat{q}_{\Phi(\boldsymbol{x})}$ is the $\Phi(\boldsymbol{x})^{th}$ entry of vector $\widehat{\boldsymbol{q}}$. In this case, we may consider $\widehat{F}(\widehat{\boldsymbol{q}})$ as a function in vector space, *i.e.*, $\widehat{F}(\cdot) : [0, 1]^{|\mathcal{X}|} \to \mathbb{R}$, and $F[q] = \widehat{F}(\widehat{\boldsymbol{q}})$. That is, we have re-represent the original functional $F : \mathcal{P} \to \mathbb{R}$ as a function $\widehat{F} : [0, 1]^{|\mathcal{X}|} \to \mathbb{R}$. The function $\widehat{F}(\cdot)$ naturally has its gradient.

**Definition 6.** *Gradient of $\widehat{F}(\cdot)$. The gradient of $\widehat{F}(\cdot) : [0, 1]^{|\mathcal{X}|} \to \mathbb{R}$ is*

$$\nabla \widehat{F}(\widehat{\boldsymbol{q}}) = \left[ \frac{\partial \widehat{F}}{\partial \widehat{q}_1}, \cdots, \frac{\partial \widehat{F}}{\partial \widehat{q}_{|\mathcal{X}|}} \right]^\top$$

We next show that no matter which bijection map is chosen, the gradient of $\widehat{F}(\cdot)$, *i.e.*, Definition 6. and the variational derivative of $F[\cdot]$, *i.e.*, Definition 3, are equivalent.

**Theorem 5.** *In the case that $\mathcal{X}$ is discrete, definition 3 is equivalent to definition 6 given any bijection $\Phi(\cdot) : \mathcal{X} \to [|\mathcal{X}|]$, i.e.,*

$$\frac{\delta F}{\delta q}(\boldsymbol{x}) = \nabla \widehat{F}(\widehat{\boldsymbol{q}})_{\Phi(\boldsymbol{x})}$$

*for any $\boldsymbol{x} \in \mathcal{X}$, where $\nabla \widehat{F}(\widehat{\boldsymbol{q}})_{\Phi(\boldsymbol{x})}$ is the $\Phi(\boldsymbol{x})^{th}$ entry of gradient vector $\nabla \widehat{F}(\widehat{\boldsymbol{q}})$.*

*Proof.* First we define an operator $\mathtt{vec}(\cdot)$ from function space over $\mathcal{X}$ to vector space $\mathbb{R}^{|\mathcal{X}|}$, so that for a function $f(\cdot) : \mathcal{X} \to \mathbb{R}$ and every $\boldsymbol{x} \in \mathcal{X}$, we have $f(\boldsymbol{x}) = \mathtt{vec}(f)_{\Phi(\boldsymbol{x})}$. That is, by using $\mathtt{vec}(\cdot)$, we store every information of $f(\cdot)$ into a vector. Therefore, by substituting $\frac{\delta F}{\delta q}(\boldsymbol{x})$ by $\nabla \widehat{F}(\widehat{\boldsymbol{q}})_{\Phi(\boldsymbol{x})}$ in definition 3, we have:

$$\sum_{\boldsymbol{x} \in \mathcal{X}} \nabla \widehat{F}(\widehat{\boldsymbol{q}})_{\Phi(\boldsymbol{x})} \phi = \mathtt{vec}(\phi)^\top \nabla \widehat{F}(\widehat{\boldsymbol{q}}) = \lim_{\epsilon \to 0} \frac{\widehat{F}(\widehat{\boldsymbol{q}} + \epsilon \mathtt{vec}(\phi)) - \widehat{F}(\widehat{\boldsymbol{q}})}{\epsilon}$$

$$= \left[ \frac{d}{d\epsilon} \widehat{F}(\widehat{\boldsymbol{q}} + \epsilon \mathtt{vec}(\phi)) \right]_{\epsilon=0} = \left[ \frac{d}{d\epsilon} \widehat{F}(\mathtt{vec}(q + \epsilon \phi)) \right]_{\epsilon=0} = \left[ \frac{dF[q + \epsilon \phi]}{d\epsilon} \right]_{\epsilon=0}$$

This shows that $\frac{\delta F}{\delta q} = \mathtt{vec}^{-1}(\nabla \widehat{F}(\widehat{\boldsymbol{q}})_{\Phi(\boldsymbol{x})})$, *i.e.*, $\frac{\delta F}{\delta q}(\boldsymbol{x}) = \nabla \widehat{F}(\widehat{\boldsymbol{q}})_{\Phi(\boldsymbol{x})}$ for any $\boldsymbol{x} \in \mathcal{X}$, where the inverse of $\mathtt{vec}(\cdot)$ exists because that the $\Phi(\cdot)$ is bijection. $\square$

Though they are equivalent in discrete settings, we can see that definition 3 is more general than definition 6, as the variational derivative can be easily extended to continuous $\mathcal{X}$. Even when $\mathcal{X}$ is discrete, it can also be used when $q$ is given by a function, with no need to store everything in a vector.

## C  Discussion on Other Projection Heuristics

One may ponder how may other heuristics perform when dealing with the $\ell_2$-norm sparse projection. For example, a two-stage thresholding approach, i.e. $i)$ running gradient descent to convergence, and then $ii)$ projection. However, it is known to be sub-optimal even for simple problems, such as least-squares with sparsity constraints. In fact, the results when using such approach on $\ell_2$-norm minimization are the same as the Greedy baseline: It $i)$ converges to the global optimum $q(\cdot)$, and then $ii)$ use greedy projection to try to minimize $F[p(\cdot)] = \|p(\cdot) - q(\cdot)\|_2^2$ subject to sparsity constraint, which has been shown to be inferior than IHT (subsection 4.1).

One may also come up with the idea of choosing the $k$ "heaviest" coordinates as support. However, in the general case, taking the $k$-heaviest coordinates of $q$ (without assuming any structure) would result into a non-valid putative solution; recall, by definition of the discrete setting, we have $n$ coordinates, each of which takes $m$ points, leading to a $m^n$ sample space. Simply taking the $k$-heaviest coordinates of that long vector would result into an intermediate representation of non-zero positions that does not correspond to a probability distribution. A variation of this approach is fine for the "vector-sparsity" special case.

## D  Proof of Theorem 1

Here, we show that the subset selection problem can be reduced to the sparse $l_2$ distribution $l_2$-norm projection problem (2). Let us first define the subset sum problem, or SSP.

**Definition 7** (Subset Sum Problem [26]). *Given a ground set of integers, $\mathcal{G} \subset \{\mathbb{Z}\}^n$, in the Subset Sum Problem we look for a non-empty subset $\mathcal{S} \subseteq \mathcal{G}$, such that the sum of all elements in $\mathcal{S}$ is zero. This is an NP-complete problem.*

Consider a Subset Sum Problem instance with a ground set $\mathcal{G}$, where $|\mathcal{G}| = n$. Let us denote its elements as $e_1, \ldots, e_n$. We reformulate the sparse distribution $\ell_2$-norm projection problem as follows. Let $\mathcal{X}$ be $n$-dimensional binary space, *i.e.*, $\mathcal{X} = \{0, 1\}^n$. For $\boldsymbol{x} \in \mathcal{X}$, let its positive positions denote a subset, *i.e.*, $\mathcal{G}_{\boldsymbol{x}} = \{e_i \mid \boldsymbol{x}_i = 1\}$. Define an $n$-dimensional function $q_k : \mathcal{X} \to \mathbb{R}$ as

$$q_k(\boldsymbol{x}) = \begin{cases} 1, & \text{if} \quad \sum_{e \in \mathcal{G}_{\boldsymbol{x}}} e = 0 \quad \text{and} \quad |\mathcal{G}_{\boldsymbol{x}}| = k, \\ 0, & \text{otherwise}, \end{cases}$$

where $k$ is a parameter.

Then, we try to find its projection to $\mathcal{D}_k$ from $k = 1$ to $k = n$. Denote $\widehat{p}_k(\cdot)$ as the optimal $\ell_2$-norm projection of $q_k(\cdot)$ to $k$-sparse distribution set $\mathcal{D}_k$. We can see that, if there is no subset $\mathcal{G}_{\boldsymbol{x}}$ with size $k$ that sum up to zero, then $q_k(\cdot)$ is zero everywhere. Denote the support of the optimal projection $\widehat{p}_k(\cdot)$ as $\mathcal{S}^\star$, and we can see that $\widehat{p}_k(\boldsymbol{x}) = \frac{1}{2^k}$ for every $\text{supp}(\boldsymbol{x}) \subseteq \mathcal{S}^\star$. That is, $\widehat{p}_k(\boldsymbol{0}) = \frac{1}{2^k}$, since $supp(\boldsymbol{0}) = \emptyset \subseteq \mathcal{S}^\star$.

If there exist a subset $\mathcal{G}_{\boldsymbol{x}'}$ with size $k$ that sum up to zero, we can see that $\widehat{p}_k(\boldsymbol{x}) = 1$ when $\boldsymbol{x} = \boldsymbol{x}'$ and $\widehat{p}_k(\boldsymbol{x}) = 0$ elsewhere. Therefore, noting that $|supp(\boldsymbol{x}')| = k$, we can check whether $\widehat{p}_k$ –*i.e.*, the optimal $\ell_2$-norm projection of $q_k(\cdot)$ to $\mathcal{D}_k$– has $\widehat{p}_k(\boldsymbol{0}) = \frac{1}{2^k}$, to know that whether there exist a subset $\mathcal{G}_{\boldsymbol{x}} \subseteq \mathcal{G}$ with size $k$ summing up to zero.

If there exist a polynomial algorithm solving the projection problem in $O(\text{poly}(n))$, then we can run it $O(n)$ times to try from $k = 1$ to $k = n$, to solve the Subset Sum Problem. Since the Subset Sum Problem is NP-hard, hence the NP-hardness of the sparse distribution $\ell_2$-norm projection problem.

## E  Proof of Theorem 2

We prove the theorem by showing that, for any algorithm we can design an example where the algorithm fails. Note that in Algorithm 1 the input of projection step is not necessarily a distribution. That is, we have to consider the input of the projection problem (2) a general function. Let $\mathcal{X}$ be $n$-dimensional binary space, *i.e.*, $\mathcal{X} = \{0, 1\}^n$. Denote an always-zero function $q_0 : \mathcal{X} \to 0$. Given an deterministic algorithm $f$, it takes in a function $q : \mathcal{X} \to \mathbb{R}$ and output a distribution $\widehat{p}(\cdot)$ with support $\mathcal{S}$, where $|\mathcal{S}| = k$. Assume the algorithm $f$ evaluates $T = O(poly(n))$ positions, denoting as $\boldsymbol{x}_1, \boldsymbol{x}_2, \ldots, \boldsymbol{x}_T$. Note that $T$ can be much less than $\binom{n}{k}$, as $\binom{n}{k}$ cannot be upper bound by any $n^c$ where $c$ is a constant. Therefore, there exist an $\boldsymbol{x}^\star$ as

$$\boldsymbol{x}^\star \in \mathcal{X} \backslash \{\boldsymbol{x}_1, \ldots, \boldsymbol{x}_T\} \quad \text{s.t.} \quad \text{supp}(\boldsymbol{x}) \neq \mathcal{S} \quad and \quad |\text{supp}(\boldsymbol{x})| = k$$

Now we construct an $n$-dimensional function $q : \mathcal{X} \to \mathbb{R}$ as

$$q(\boldsymbol{x}) = \left\{ \begin{array}{ll} 1 + \delta, & \text{if} \quad \boldsymbol{x} = \boldsymbol{x}^\star \\ 0, & \text{otherwise} \end{array} \right. ,$$

where $\delta > 0$. We input the constructed $q$ to the deterministic algorithm $f$. Note that the value of positions it evaluates do not have any differences compared to those when $q_0$ is inputed, *i.e.*, $q(\boldsymbol{x}_i) = q_0(\boldsymbol{x}_i) = 0$ for every $i \in [T]$. As $f$ is deterministic, the output solution is still $\widehat{p}$ with support $\mathcal{S}$. As a result, $q(\boldsymbol{x}) = 0$ for every $\boldsymbol{x} \in \mathcal{X}_{\mathcal{S}}$. Denoting $\widehat{p}_{\mathcal{S}}$ as the optimal $\ell_2$-norm projection of $q$ to $\mathcal{P}_{\mathcal{S}}$, we have $\|q(\cdot) - \widehat{p}(\cdot)\|_2^2 \geq \|q(\cdot) - \widehat{p}_{\mathcal{S}}\|_2^2 = 1/|\mathcal{X}_{\mathcal{S}}| + (1 + \delta)^2$.

Noting that the optimal projection is

$$p^\star(\boldsymbol{x}) = \left\{ \begin{array}{ll} 1, & \text{if} \quad \boldsymbol{x} = \boldsymbol{x}^\star \\ 0, & \text{otherwise} \end{array} \right. ,$$

we can see the optimal $\ell_2$-norm distance is $\|q(\cdot) - p^\star(\cdot)\|_2^2 = \delta^2$. Therefore, the approximation rate of algorithm $f$ on this input $q$ is

$$\varphi = \frac{\|q - \widehat{p}\|_2^2}{\|q - p^\star\|_2^2} - 1 \geq \frac{\frac{1}{|\mathcal{X}_{\mathcal{S}}|} + (1 + \delta)^2}{\delta^2} - 1 \geq \frac{\frac{1}{|\mathcal{X}_{\mathcal{S}}|} + 1}{\delta^2} - 1$$

As $\delta$ can be arbitrarily close to 0, the approximation ratio $\varphi$ can not be upper bounded.

## F   Proof of Theorem 3

*Proof.* First, we quantify the influence of the gradient step. For any support $\mathcal{S} \subset [n]$, let $\hat{q}_{\mathcal{S}}$ be the optimal projection of $q = p - \mu \frac{\delta F}{\delta p}$ to sparsity domain $\mathcal{P}_{\mathcal{S}}$, and let $\hat{p}_{\mathcal{S}}$ be the optimal projection of $p$ to sparsity domain $\mathcal{P}_{\mathcal{S}}$. Then, we have

$$\|\hat{q}_{\mathcal{S}} - q\|_2 = \left\| \hat{q}_{\mathcal{S}} - p + \mu \frac{\delta F}{\delta p} \right\|_2 \geq \|\hat{q}_{\mathcal{S}} - p\|_2 - \mu L \geq \|\hat{p}_{\mathcal{S}} - p\|_2 - \mu L$$

Its upper bound is

$$\|\hat{q}_{\mathcal{S}} - q\|_2 \leq \|\hat{p}_{\mathcal{S}} - q\|_2 = \left\| \hat{p}_{\mathcal{S}} - p + \mu \frac{\delta F}{\delta p} \right\|_2 \leq \|\hat{p}_{\mathcal{S}} - p\|_2 + \mu L$$

That is,

$$\|\hat{p}_{\mathcal{S}} - p\|_2 + \mu L \geq \|\hat{q}_{\mathcal{S}} - q\|_2 \geq \|\hat{p}_{\mathcal{S}} - p\|_2 - \mu L \tag{5}$$

Nest, consider a support $\mathcal{S} \subset \mathcal{S}'$, where $\mathcal{S}' = supp(p)$. We use the greedy procedure to add one element $e \in [n] \backslash \mathcal{S}$ to $\mathcal{S}$. It is to find

$$e \in \arg \min_{i \in [n] \backslash \mathcal{S}} \|\hat{q}_{\mathcal{S} \cup i} - q\|_2$$

Define $\theta$ as a parameter describing how much better if we choose support $\mathcal{S} \subset \mathcal{S}'$ to project than choosing other supports.

$$\theta = \min_{\mathcal{S} : \mathcal{S} \subset \mathcal{S}'} \left( \min_{i \in [n] \backslash \mathcal{S}', j \in \mathcal{S}' \backslash \mathcal{S}} \|\hat{q}_{\mathcal{S} \cup i} - q\|_2 - \|\hat{q}_{\mathcal{S} \cup j} - q\|_2 \right)$$

As we can see, the greater $\theta$ is, the more possible for the greedy method to finally find $\mathcal{S}'$. Moreover, if $\theta > 0$, then the greedy procedure finds exactly $\mathcal{S}'$. By using inequality (5), we have

$$\theta > \min_{\mathcal{S} : \mathcal{S} \subset \mathcal{S}'} \left( \min_{i \in [n] \backslash \mathcal{S}', j \in \mathcal{S}' \backslash \mathcal{S}} \|\hat{p}_{\mathcal{S} \cup i} - p\|_2 - \|\hat{p}_{\mathcal{S} \cup j} - p\|_2 \right) - 2\mu L$$

Therefore, if we have

$$2\mu L < \min_{\mathcal{S} : \mathcal{S} \subset \mathcal{S}'} \left( \min_{i \in [n] \backslash \mathcal{S}', j \in \mathcal{S}' \backslash \mathcal{S}} \|\hat{p}_{\mathcal{S} \cup i} - p\|_2 - \|\hat{p}_{\mathcal{S} \cup j} - p\|_2 \right), \tag{6}$$

we can guarantee $\theta > 0$, which means the greedy method finds exactly $\mathcal{S}'$.

Next, we analyze when is inequality (6) achievable for enough small step size $\mu > 0$, *i.e.*, $\|\hat{p}_{\mathcal{S}\cup i} - p\|_2 - \|\hat{p}_{\mathcal{S}\cup j} - p\|_2 > 0$ in inequality (6).

First, let us calculate $\|\hat{p}_{\mathcal{S}\cup i} - p\|_2^2$ for any $i \in [n]\backslash\mathcal{S}'$.

$$
\begin{aligned}
\|\hat{p}_{\mathcal{S}\cup i} - p\|_2^2 &= \sum_{supp(\boldsymbol{x})\leq|\mathcal{S}\cup i|} (\hat{p}_{\mathcal{S}\cup i}(\boldsymbol{x}) - p(\boldsymbol{x}))^2 + \sum_{supp(\boldsymbol{x})>|\mathcal{S}\cup i|} (\hat{p}_{\mathcal{S}\cup i}(\boldsymbol{x}) - p(\boldsymbol{x}))^2 \\
&= \sum_{supp(\boldsymbol{x})\leq|\mathcal{S}\cup i|} (\hat{p}_{\mathcal{S}\cup i}(\boldsymbol{x}) - p(\boldsymbol{x}))^2 + \sum_{supp(\boldsymbol{x})>|\mathcal{S}\cup i|} p(\boldsymbol{x})^2 \\
&= \sum_{\boldsymbol{x}\in\mathcal{X}_{\mathcal{S}\cup i}} (\hat{p}_{\mathcal{S}\cup i}(\boldsymbol{x}) - p(\boldsymbol{x}))^2 + \sum_{supp(\boldsymbol{x})\leq|\mathcal{S}\cup i|,\boldsymbol{x}\notin\mathcal{X}_{\mathcal{S}\cup i}} (\hat{p}_{\mathcal{S}\cup i}(\boldsymbol{x}) - p(\boldsymbol{x}))^2 \\
&\quad + \sum_{supp(\boldsymbol{x})>|\mathcal{S}\cup i|} p(\boldsymbol{x})^2 \\
&= \sum_{\boldsymbol{x}\in\mathcal{X}_{\mathcal{S}\cup i}} (\hat{p}_{\mathcal{S}\cup i}(\boldsymbol{x}) - p(\boldsymbol{x}))^2 + \sum_{supp(\boldsymbol{x})\leq|\mathcal{S}\cup i|,\boldsymbol{x}\notin\mathcal{X}_{\mathcal{S}\cup i}} p(\boldsymbol{x})^2 + \sum_{supp(\boldsymbol{x})>|\mathcal{S}\cup i|} p(\boldsymbol{x})^2 \\
&= \sum_{\boldsymbol{x}\in\mathcal{X}_{\mathcal{S}\cup i}} (\hat{p}_{\mathcal{S}\cup i}(\boldsymbol{x}) - p(\boldsymbol{x}))^2 - \sum_{\boldsymbol{x}\in\mathcal{X}_{\mathcal{S}\cup i}} p(\boldsymbol{x})^2 + \sum_{\boldsymbol{x}\in\mathcal{X}} p(\boldsymbol{x})^2 \quad (7)
\end{aligned}
$$

Noting that $p : \mathcal{X} \to \mathbb{R}_+$ is a distribution, we can explicitly find $\hat{p}_{\mathcal{S}\cup i}$, as shown in the main text, then

$$
\sum_{\boldsymbol{x}\in\mathcal{X}_{\mathcal{S}\cup i}} (\hat{p}_{\mathcal{S}\cup i}(\boldsymbol{x}) - p(\boldsymbol{x}))^2 = \frac{(1 - \sum_{\boldsymbol{x}\in\mathcal{X}_{\mathcal{S}\cup i}} p(\boldsymbol{x}))^2}{|\mathcal{X}_{\mathcal{S}\cup i}|}
$$

Substituting it into equation (7), we have

$$
\|\hat{p}_{\mathcal{S}\cup i} - p\|_2^2 = \frac{(1 - \sum_{\boldsymbol{x}\in\mathcal{X}_{\mathcal{S}\cup i}} p(\boldsymbol{x}))^2}{|\mathcal{X}_{\mathcal{S}\cup i}|} - \sum_{\boldsymbol{x}\in\mathcal{X}_{\mathcal{S}\cup i}} p(\boldsymbol{x})^2 + \sum_{\boldsymbol{x}\in\mathcal{X}} p(\boldsymbol{x})^2 \quad (8)
$$

We can see that the derivation of equation (8) also holds for $j \in \mathcal{S}'\backslash\mathcal{S}$, which means

$$
\|\hat{p}_{\mathcal{S}\cup j} - p\|_2^2 = \frac{(1 - \sum_{\boldsymbol{x}\in\mathcal{X}_{\mathcal{S}\cup j}} p(\boldsymbol{x}))^2}{|\mathcal{X}_{\mathcal{S}\cup j}|} - \sum_{\boldsymbol{x}\in\mathcal{X}_{\mathcal{S}\cup j}} p(\boldsymbol{x})^2 + \sum_{\boldsymbol{x}\in\mathcal{X}} p(\boldsymbol{x})^2
$$

In our discrete setting, we have $|\mathcal{X}_{\mathcal{S}\cup i}| = |\mathcal{X}_{\mathcal{S}\cup j}|$.

Therefore,

$$
\begin{aligned}
&\|\hat{p}_{\mathcal{S}\cup i} - p\|_2^2 - \|\hat{p}_{\mathcal{S}\cup j} - p\|_2^2 \\
&= \frac{(1 - \sum_{\boldsymbol{x}\in\mathcal{X}_{\mathcal{S}\cup i}} p(\boldsymbol{x}))^2 - (1 - \sum_{\boldsymbol{x}\in\mathcal{X}_{\mathcal{S}\cup j}} p(\boldsymbol{x}))^2}{|\mathcal{X}_{\mathcal{S}\cup j}|} + \left(\sum_{\boldsymbol{x}\in\mathcal{X}_{\mathcal{S}\cup j}} p(\boldsymbol{x})^2 - \sum_{\boldsymbol{x}\in\mathcal{X}_{\mathcal{S}\cup i}} p(\boldsymbol{x})^2\right) \\
&= \frac{\left(2 - \sum_{\boldsymbol{x}\in\mathcal{X}_{\mathcal{S}\cup i}} p(\boldsymbol{x}) - \sum_{\boldsymbol{x}\in\mathcal{X}_{\mathcal{S}\cup j}} p(\boldsymbol{x})\right)\left(\sum_{\boldsymbol{x}\in\mathcal{X}_{\mathcal{S}\cup j}} p(\boldsymbol{x}) - \sum_{\boldsymbol{x}\in\mathcal{X}_{\mathcal{S}\cup i}} p(\boldsymbol{x})\right)}{|\mathcal{X}_{\mathcal{S}\cup j}|} \\
&\quad + \left(\sum_{\boldsymbol{x}\in\mathcal{X}_{\mathcal{S}\cup j}} p(\boldsymbol{x})^2 - \sum_{\boldsymbol{x}\in\mathcal{X}_{\mathcal{S}\cup i}} p(\boldsymbol{x})^2\right)
\end{aligned}
$$

Noting that $p$ is a $k$-sparse distribution with support $\mathcal{S}'$, we can see that for $i \in [n]\backslash\mathcal{S}'$, $\{\mathcal{X}_{\mathcal{S}\cup i}\backslash\mathcal{X}_\mathcal{S}\} \cap \mathcal{X}_{\mathcal{S}'} = \emptyset$. Therefore, $p(\boldsymbol{x}) = 0$ where $\boldsymbol{x} \in \mathcal{X}_{\mathcal{S}\cup i}\backslash\mathcal{X}_\mathcal{S}$. Hence, we can simplify the previous equation as

$$
\begin{aligned}
&\|\hat{p}_{\mathcal{S}\cup i} - p\|_2^2 - \|\hat{p}_{\mathcal{S}\cup j} - p\|_2^2 \\
&= \frac{\left(2 - \sum_{\boldsymbol{x}\in\mathcal{X}_{\mathcal{S}\cup i}} p(\boldsymbol{x}) - \sum_{\boldsymbol{x}\in\mathcal{X}_{\mathcal{S}\cup j}} p(\boldsymbol{x})\right)\left(\sum_{\boldsymbol{x}\in\mathcal{X}_{\mathcal{S}\cup j}\backslash\mathcal{X}_\mathcal{S}} p(\boldsymbol{x})\right)}{|\mathcal{X}_{\mathcal{S}\cup j}|} + \left(\sum_{\boldsymbol{x}\in\mathcal{X}_{\mathcal{S}\cup j}\backslash\mathcal{X}_\mathcal{S}} p(\boldsymbol{x})^2\right)
\end{aligned}
$$

As we can see, if there exist $x \in \mathcal{X}_{\mathcal{S}\cup j}\backslash\mathcal{X}_{\mathcal{S}}$ for all $\mathcal{S} \subset \mathcal{S}', i \in [n]\backslash\mathcal{S}', j \in \mathcal{S}'\backslash\mathcal{S}$, such that $p(x) > 0$, then $\|\hat{p}_{\mathcal{S}\cup i} - p\|_2^2 - \|\hat{p}_{\mathcal{S}\cup j} - p\|_2^2 > 0$ for all the $\mathcal{S}, i, j$. That is, conceptually, there are enough positions $x \in \mathcal{X}_{\mathcal{S}'}$ where $p(x) > 0$. And this condition leads us to the following wanted inequality.

$$\min_{\mathcal{S}:\mathcal{S}\subset\mathcal{S}'} \left( \min_{i\in[n]\backslash\mathcal{S}',j\in\mathcal{S}'\backslash\mathcal{S}} \|\hat{p}_{\mathcal{S}\cup i} - p\|_2 - \|\hat{p}_{\mathcal{S}\cup j} - p\|_2 \right) > 0. \tag{9}$$

Hence, under such conditions, *i.e.*, inequality (9) and inequality (6) hold, the greedy method is guaranteed to find exactly $\mathcal{S}'$.

$\square$

# G  Proof of Theorem 4

*Proof.* Considering iteration $t$ and $t + 1$ as in algorithm 1. We drop parentheses for clarity. Applying RSS property we have:

$$F[p^{t+1}] - F[p^t] \leq \left\langle \frac{\delta F}{\delta p^t}, p^{t+1} - p^t \right\rangle + \frac{\beta}{2}\|p^{t+1} - p^t\|_2^2$$

$$= \frac{1}{\mu}\langle p^t - q^{t+1}, p^{t+1} - p^t \rangle + \frac{\beta}{2}\|p^{t+1} - p^t\|_2^2$$

Setting step size $\mu = 1/\beta$, and then complete the square:

$$F[p^{t+1}] - F[p^t] \leq \frac{\beta}{2}(\|p^{t+1} - q^{t+1}\|_2^2 - \|p^t - q^{t+1}\|_2^2)$$

$$\leq \frac{\beta}{2}\left[(1+\phi)\|p^* - q^{t+1}\|_2^2 - \|p^t - q^{t+1}\|_2^2\right]$$

where the inequality is due to approximate projection. Now by adding and subtracting $p^t$ in $\|p^* - q^{t+1}\|_2^2$ on the right hand side, we have:

$$\frac{\beta}{2}\left[(1+\phi)\left(\|p^* - p^t\|_2^2 + \|p^t - q^{t+1}\|_2^2 + 2\langle p^\star - p^t, p^t - q^{t+1}\rangle\right) - \|p^t - q^{t+1}\|_2^2\right]$$

$$= \frac{\beta}{2}\left[(1+\phi)\|p^* - p^t\|_2^2 + \phi\|p^t - q^{t+1}\|_2^2 + 2(1+\phi)\langle p^\star - p^t, p^t - q^{t+1}\rangle\right]$$

$$= \frac{\beta}{2}(1+\phi)\|p^* - p^t\|_2^2 + \frac{\phi}{2\beta}\|\frac{\delta F}{\delta p^t}\|_2^2 + (1+\phi)\langle p^\star - p^t, \frac{\delta F}{\delta p^t}\rangle$$

Applying the Lipschitz condition 2, we have:

$$F[p^{t+1}] - F[p^t] \leq \frac{\beta}{2}(1+\phi)\|p^* - p^t\|_2^2 + \frac{\phi L^2}{2\beta} + (1+\phi)\langle p^\star - p^t, \frac{\delta F}{\delta p^t}\rangle \tag{10}$$

To bound the last inner product in (10), we need RSC property:

$$\langle p^\star - p^t, \frac{\delta F}{\delta p^t}\rangle \leq F[p^\star] - F[p^t] - \frac{\alpha}{2}\|p^t - p^\star\|_2^2$$

Apply it to relax the inner product term, we have (10):

$$\leq \frac{\beta - \alpha}{2}(1+\phi)\|p^\star - p^t\|_2^2 + \frac{\phi L^2}{2\beta} + (1+\phi)[F[p^\star] - F[p^t]] \tag{11}$$

Next we find the relation between $F[p^\star] - F[p^t]$ and $\|p^\star - p^t\|_2^2$ by RSC:

$$F[p^t] - F[p^\star] \geq \langle \frac{\delta F}{\delta p^\star}, p^t - p^\star \rangle + \frac{\alpha}{2}\|p^t - p^\star\|_2^2 \geq \frac{\alpha}{2}\|p^t - p^\star\|_2^2 - \|\frac{\delta F}{\delta p^\star}\|_2 \cdot \|p^t - p^\star\|_2$$

$$\geq \frac{\alpha}{2}\|p^t - p^\star\|_2^2 - L\|p^t - p^\star\|_2 = \frac{\alpha}{2}\left[\|p^t - p^\star\|_2 - \frac{L}{\alpha}\right]^2 - \frac{L^2}{2\alpha}$$

where the second inequality is by Cauchy–Schwarz inequality. When $\|p^t - p^\star\|_2 \leq \frac{L}{2\alpha}$, we have

$$F[p^t] - F[p^\star] \geq \frac{\alpha}{2}\|p^t - p^\star\|_2^2 - \frac{L^2}{2\alpha} \tag{12}$$

Apply (12) to (11) to convert $\|p^t - p^\star\|_2$ to $F[p^t] - F[p^\star]$, we have (11)

$$\leq (1+\phi)(2 - \frac{\beta}{\alpha})[F[p^\star] - F[p^t]] + \left( \frac{\phi}{2\beta} + (1+\phi)\frac{\beta - \alpha}{2\alpha^2} \right) L^2 \tag{13}$$

We denote the last term in (13) as $c_1$, and rearrange the equation, we have

$$F[p^{t+1}] - F[p^\star] \leq (1 - (1+\phi)(2 - \beta/\alpha)) \left[ F[p^t] - F[p^\star] \right] + c_1$$

$$F[p^{t+1}] - F[p^\star] - c \leq (1 - (1+\phi)(2 - \beta/\alpha)) \left[ F[p^t] - F[p^\star] - c \right] \tag{14}$$

where

$$c = \frac{c_1}{(1+\phi)(2 - \beta/\alpha)} = \frac{\left( \phi/(2\beta) + (1+\phi)(\beta - \alpha)/(2\alpha^2) \right) L^2}{(1+\phi)(2 - \beta/\alpha)}$$

From (14), we can see that if $0 < (1+\phi)(2 - \beta/\alpha) < 1$, or $2 - 1/(1+\phi) < \beta/\alpha < 2$, IHT is guaranteed to converge to $F[p^\star] + c$ linearly. The smaller $\phi$ is and the closer is $\beta$ to $\alpha$, the smaller $c$ is.

$\square$

# H    Aditional Experimental details

**Model Compression / Compressed sensing**

We apply our IHT to the task of expectation-preserving distribution compression, useful for efficiently storing large probability tables. Given a distribution $p(\cdot)$, our goal is to construct a sparse approximation $q(\cdot)$, such that $q(\cdot)$ approximately preserves expectations with respect to $p(\cdot)$. Interestingly, this model compression problem is equivalent to compressed sensing, but with the distributional constraints. We consider the vector sparsity in this experiment, *i.e.*, the distribution $q$ is represented as a long vector in space $[0, 1]^n$, as described in Appendix A. The problem setting we use in this experiment is to minimize $\|Aq - Ap\|_2^2$ subject to the vector distribution $k$-sparsity constraint of $q$, *i.e.*, $\|q\|_0 \leq k$ and $\sum_{i \in [n]} q_i = 1$. We first train the algorithms to minimize $\|Aq - Ap\|_2^2$ and then test their error on $\|Bq - Bp\|_2^2$, where $A, B$ are randomly drawn from normal distribution $\mathcal{N}(0, 1)$, and $p \in [0, 1]^n$ is a distribution generated from data.

We use real-world data: *Total Charges in 2012 Base Data 1, Hospital Discharge Data Public Use Data File*[1], which contains 740817 records. We set 10000 bins with bin size of 1000, to converge the data into a histogram, and hence the distribution $p \in [0, 1]^{10000}$. Note that $p$ is already sparse. We set the dimension of $A, B$ to $500 \times 10000$.

We compare IHT with two baselines, *i.e.*, Lasso and Random. Note that in vector-sparsity setting, the sparse $l_2$ distribution projection can be done optimally, since it becomes essentially a projection to a simplex, as we discussed in the main paper. The Lasso baseline is to minimize $\|Aq - Ap\|_2^2 + \gamma\|q\|_1$ and then project its solution to the $k$-sparse distribution domain. The Random baseline is to randomly generate $T$ $k$-sparse distribution, and simply choose the best, where $T$ is the iteration number of IHT. Note that though the Greedy algorithm can work on this setting theoretically, it is too time costly to compare with the previously mentioned three algorithms.

As both training matrix $A$ and testing matrix $B$ are randomly generated, we use 10 different $A$ for which we train the algorithms for 10 times, and after each training process we generate 20 test matrices $B$ to test the error of each algorithms.

(a) IHT converges fast ($k = 200$)

(b) Error Comparison with different sparsity $k$

Figure 4: Real-data Experiments

Figure 4 (a) gives the convergence result when we set sparsity level $k = 200$. The IHT Train Error shows the training error of IHT at each iterations. IHT Error, Lasso Error and Random Error are testing errors of the three algorithms after training. We can see the promising results of IHT which outperforms other baselines. In Figure 4 (b), we test the three algorithms on different sparsity level $k = 100 \cdots 500$. IHT, Lasso and Random are testing errors of the three algorithms after training. Our results verify that IHT does the best regardless of sparsity level $k$.

**Digits data: Dataset compression**

We study representative prototype selection for the Digits data [31], which contains 7291 training and 2007 test examples of handwritten grayscale images. Prototypes are representative examples chosen from the data, in order to achieve dataset compression, while preserving certain desirable properties. In this experiment, our goal is to achieve compression to speed up nearest neighbor classification on unseen data as the quality measure of the selected prototypes. To this end, we embed the data using the RBF kernel $\exp(\gamma |\mathbf{x}_i - \mathbf{x}_i|^2)$, where the parameter $\gamma$ is set using cross validation, and use the Maximum Mean Discrepancy (MMD) between the discrete data distribution in the embedded space, with the sparse data distribution representing the selected samples as our cost function for IHT. For two densities $p$ and $p$, we can write $\text{MMD}^2 = E_{x,y \sim p} K(x,y) - 2E_{x \sim p, y \sim q} K(x,y) + E_{x \sim q, y \sim q} K(x,y)$, where $K(\cdot, \cdot)$ is the RBF kernel function. After the prototypes are selected, we evaluate the $0/1$ classification error with 1 Nearest Neighbor on the test data using only the selected prototypes.

We compare two forward selection greedy variants (Local Greedy and Global Greedy) proposed by [32] and the means algorithm (labeled as PS) proposed by [33], both state of the art. The results are presented in Figure 3(b). We see that IHT performs better than the baselines across different number of selected prototypes, especially when the number of prototypes is smaller.

## Footnotes

[1] https://www.dshs.state.tx.us/THCIC/Hospitals/Download.shtm