[Reviews · NeurIPS 2019]

Reviewer 1



Post-rebuttal: I have downgraded my overall score to 7. I am troubled by the lack of motivation (and that in the rebuttal, the authors defer more discussion of model compression to future work). Also, I'd have liked to see in the rebuttal more details about the "more comprehensive discussion" regarding alternate algorithms. ------------------ ORIGINALITY =========== Motivated by previous work on modeling priors for functional neuroimaging data as sparse distributions, this paper studies the problem of learning a sparse distribution that minimizes a convex loss function. Apparently, this is the first work that studies this problem for general loss functions. The goal of the work is to adapt the well-known IHT algorithm, a form of projected gradient descent, to this problem. Again, as far as I can see, this approach is original to this work. The mathematical techniques are based on standard approaches and are not very novel in my opinion. QUALITY & CLARITY ================= The paper is mostly a pleasure to read. It tackles head-on the stated goal of investigating the performance of IHT, identifies a computational barrier to effient implementation, and then gives general conditions (strong convexity, smoothness, lipschitzness) that enable a greedy heuristic to be correct. The proofs are solid but not very complicated. The experimental section is brief and a bit unsatisfactory, as the comparisons are against simple baselines. Why not try some others? * Analogously to basic thresholding, first solve the problem without the sparsity constraint, then take the heaviest k coordinates, and solve the problem restricted to distributions with support on just those coordinates. * Consider an alternate projection algorithm in IHT, where you take the heaviest k coordinates of q and find a distribution supported on those k coordinates that is closest to q. Also, it would be interesting to check whether in the experiments, the support set changes during the iterations. Why not try fixing the support set after the first (or few) iterations in IHT and doing the exact projection to that set thereafter? SIGNIFICANCE ============= To the extent that optimization over sparse distributions is an important problem, the contributions in this paper are very significant and relevant to the NeurIPS community. However, I feel the authors should do a better job with the motivation. Is it just [5] that shows the utility of modeling distributions as sparse? Why do they not discuss the model compression problem that's used for the experiments?

Reviewer 2



The paper studies Iterative Hard Thresholding (IHT) in distribution space. IHT algorithms have been studied before. This work aims at kind of lifting the solutions provided by IHT to distribution space, i.e., to distributions with (usually many) `hard' zeros on the discrete space they are defined on. The overall approach is defined and investigated for relatively general functionals F[.]. The definition of the general framework is an achievement by itself to my mind. I also like the authors showing what can be done and what can not be done in terms of complexity. Conditions for functionals are provided and convergence results are obtained. The proofs are partly long but necessary for this domain of IHT, I believe. Sometime (e.g. 211-214) restrictions are imposed that the authors say can be made more general to be practical. This puzzles the reader. Also some claims are not supported by the theoretical results. For instance (ll198-202), the authors say that only "extreme examples" are hard to solve. But we "extreme examples" is not really defined. If one is unlucky, many real-world examples may fall under the "extreme examples" label. If not the authors should explain. In general, however, the theory part is solid and interesting. The general research direction is also relevant, as distributions with 'hard zeros' are relevant. This is btw not only true if considering compressive sensing type applications. There has been interest, e.g., in distributions with hard zeros for spike-and-slab sparse coding (Goodfellow et al., TPAMI 2012; Sheikh et al., JMLR 2014) or even for neuroscience applications (Shelton et al., NIPS 2011; Shivkumar et al., NeurIPS 2018). In this respect it would be interesting to optimize a functional for the free energy / ELBO using IHT (has to take the entropy of q into account). On the downside, the paper shows a gap between theory and numerical evaluation. Of course, it is always difficult to relate general theoretical results to concrete experiments even if they are intended as a proof of concept. But for the reader it is particularly difficult to gauge the relevance of the theoretical results (e.g., convergence rates, properties of the functional etc) to the shown and to the potential applications of the approach. The experimental section does too little to link previous properties to what is shown in the experiments. There are also open questions: In lines 277 to 280 the smaller variance of IHT compared to the other algorithms is stated as an advantage. However, if one can efficiently compute or estimate the obtained objective, then one could pick the best, e.g. of a bunch of `Greedy' runs. A large variance would then be an advantage. To turn the argument around: could one make the IHT more stochastic? And is it true that there is not variance in 20 runs obtained because the algorithm is deterministic? What about different starting conditions? After rebuttal: I do not have the feeling that all my questions were understood correctly but many were.

Reviewer 3



The main contribution is to provide with an algorithm to learn sparse distributions and would benefit by improving the way the methods are explained and the clarity is identified. In particular: - Some statements could be commented more generally, for instance " Interestingly, $Dk included in Dk' included in P$ in general." (l.79), in particular if they are novel. Explain what the "QM-AM inequality" is. - The resulting algorithm provides a further evidence that greedy algorithms are quite powerful at solving NP-hard problems, yet this statement is not completely falsifiable (are there other alternatives to using greedy methods?) and the novelty of this particular application to distributions should be justified. Indeed, it seems that most theoretical results come from extending this functional setting from vector sparsity (see supp l.408 for instance). As such, clarify the novelty of your results.

[Author Response · NeurIPS 2019]

We would like to thank all of the reviewers for their valuable time and their constructive comments. In what follows, we would like to address all concerns raised.

**Reviewer 1:** We will incorporate the proposed minor corrections in the final version of the paper. We thank the reviewer for proposing the interesting comparisons, and we will include more comprehensive discussion in the final paper. Below please find some details regarding the proposed methods.

- The two-stage approach, i.e., $i)$ running gradient descent to convergence, and then $ii)$ projection onto sparsity set, is known to be sub-optimal even for simple problems such as least-squares with sparsity constraints. In fact, the results when using such approach on $\ell_2$-norm minimization are the same as the Greedy baseline: It $i)$ converges to the global optimum $q(\cdot)$, and then $ii)$ use greedy projection to try to minimize $F[p(\cdot)] = \|p(\cdot) - q(\cdot)\|_2^2$ subject to sparsity constraint, which has been shown to be inferior than IHT (subsection 4.1).
- In the general case, taking the $k$-heaviest coordinates of $q$ (without assuming any structure) would result into a non-valid putative solution; recall, by definition of the discrete setting, we have $n$ coordinates, each of which takes $m$ points, leading to a $m^n$ sample space. Simply taking the $k$-heaviest coordinates of that long vector would result into an intermediate representation of non-zero positions that does not correspond to a probability distribution. A variation of this approach is fine for the "vector-sparsity" special case.

On whether support set changes during iterations, we observe that in experiments (subsection 4.1) IHT changes support, on average, 36.1 times for $\ell_2$-norm minimization and 6.9 times for KL minimization. We also conduct experiments on fixing the support after some iterations: IHT on $\ell_2$-norm minimization (subsection 4.1) after 400 iterations, fixing supports after 1, 5, 10 and 15 support sets change, give average results of 0.0026, 0.0020, 0.0018, 0.0016, respectively.

Regarding motivation, model compression is an exciting immediate application of our proposed approach, especially since our general approach may be flexibly applied to specialized problem-specific losses. We are currently investigating extensions of the current work to model/policy compression for reinforcement learning, where the loss can be constructed to preserve post-compression expected reward; but the details of that approach deserve a different publication.

**Reviewer 2:** We thank the reviewer for the supportive and constructive review. Regarding the comment in lines 211-214, we chose not to include this detail as extending convergence analyses from global to local strong convexity is considered fairly standard; see e.g., [Agarwal2010].

Regarding the comment in lines 198-202, we apologize for any confusion. This argument is a standard one in NP-hardness proofs: proving the existence of corner cases is sufficient to show the hardness of the problem. Note that even if *empirically* we observe that common cases are easy to solve, this does not guarantee this is the norm. Showing that with high-probability the corner cases rarely appear is an interesting question by itself.

Regarding variance in experiments, we have observed high variance is not enough for the algorithm to get "lucky". In fact, we observe that the **best** performance of Greedy is often worse than the **worst** performance of IHT. Low variance is especially desirable when the algorithm is only applied a few times to save computation, as in large discrete optimization problems. We believe that this makes IHT preferable in practice.

We also thank the reviewer for the suggestions on free energy / ELBO using IHT; we will consider these as future work.

**Reviewer 3:** We thank the reviewer for the constructive comments. We are happy to fully describe the structure nesting i.e. why $\mathcal{D}_k$ is included in $\mathcal{D}'_k$ in general. QM-AM stands for Quadratic Mean-Arithmetic Mean inequality.

Unfortunately, we are unaware of methods that are strictly better than greedy $\ell_2$-norm projection. However, trading time for performance may be an interesting topic for future work. Regarding the novelty, although we unveil a relationship between the functional derivative and the gradient (l. 408), they are different in many aspects. The $\ell_2$-norm projection for vector sparsity can be done optimally, but we have shown that $\ell_2$-norm projection for the general case is computationally hard in general (Theorems 1 & 2). Despite the provable hardness, we provide some intuition for why greedy projection is effective in our main Algorithm (Theorem 3). These are challenges of extending vector optimization to general distribution optimization. We also note that the convergence analysis is for the functional setting, not the vector setting (Theorem 4), which is not only more general, but also paves the way for future work on continuous distributions.

Regarding **why** greedy seems effective in practice, we have provided the intuition and its supporting theorem in section 3.4. Regarding the greedy projection (Algorithm 2), it is also possible to use KL divergence as distance metric, but our convergence analysis (Theorem 4) suggests that a good projection in $\ell_2$-norm (Definition 5) is preferable.

We will improve the presentation as suggested in the final version of the paper.

[Agarwal2010] Agarwal, Alekh and Negahban, Sahand and Wainwright, Martin J, "*Fast global convergence rates of gradient methods for high-dimensional statistical recovery*", Advances in Neural Information Processing Systems, 2010.

[Meta-Review · NeurIPS 2019]

This paper studies the use of iterative hard threshholding for stuctured estimation problems. Reviewers reached a clear consensus. There are a few outstanding issues that should be considered for a final version.